# FoxO3 Modulates Circadian Rhythms in Neural Stem Cells

**DOI:** 10.3390/ijms241713662

**Published:** 2023-09-04

**Authors:** Swip Draijer, Raissa Timmerman, Jesse Pannekeet, Alexandra van Harten, Elham Aida Farshadi, Julius Kemmer, Demy van Gilst, Inês Chaves, Marco F. M. Hoekman

**Affiliations:** 1Swammerdam Institute of Life Sciences, University of Amsterdam, 1018 WB Amsterdam, The Netherlandsm.f.m.hoekman@uva.nl (M.F.M.H.); 2Department of Molecular Genetics, Erasmus University Medical Center, 3000 CA Rotterdam, The Netherlands

**Keywords:** circadian rhythms, FoxO3, neural stem cell, liver, metabolism, cell cycle

## Abstract

Both FoxO transcription factors and the circadian clock act on the interface of metabolism and cell cycle regulation and are important regulators of cellular stress and stem cell homeostasis. Importantly, FoxO3 preserves the adult neural stem cell population by regulating cell cycle and cellular metabolism and has been shown to regulate circadian rhythms in the liver. However, whether FoxO3 is a regulator of circadian rhythms in neural stem cells remains unknown. Here, we show that loss of FoxO3 disrupts circadian rhythmicity in cultures of neural stem cells, an effect that is mediated via regulation of *Clock* transcriptional levels. Using Rev-Erbα-VNP as a reporter, we then demonstrate that loss of FoxO3 does not disrupt circadian rhythmicity at the single cell level. A meta-analysis of published data revealed dynamic co-occupancy of multiple circadian clock components within FoxO3 regulatory regions, indicating that FoxO3 is a Clock-controlled gene. Finally, we examined proliferation in the hippocampus of FoxO3-deficient mice and found that loss of FoxO3 delayed the circadian phase of hippocampal proliferation, indicating that FoxO3 regulates correct timing of NSC proliferation. Taken together, our data suggest that FoxO3 is an integral part of circadian regulation of neural stem cell homeostasis.

## 1. Introduction

The circadian clock is an evolutionary conserved oscillator present in most organisms generating oscillations in transcription and translation and regulating timing of cellular processes during day and night [1,2]. This is the result of the various interlocked transcription–translation feedback loops that constitute the molecular circadian clock, with a cycle spanning approximately 24 h [3,4]. Important molecular components of the circadian clock are the bHLH transcription factors CLOCK, its paralogue NPAS2, and BMAL1. The CLOCK–BMAL1—or NPAS2–BMAL1—complexes bind to E-box regulatory elements and promote expression of target genes, including clock genes PER1, PER2, PER3, CRY1 and CRY2. The latter then repress CLOCK–BMAL1 activity, resulting in rhythmic expression. Moreover, CLOCK–BMAL1 activates expression of nuclear receptors REV-ERBα and REV-ERBβ, which repress expression of *Bmal1* and *Clock* [3,4]. The suprachiasmatic nucleus (SCN) located in the brain is the light-entrainable master regulator that synchronizes the circadian clock in all peripheral tissues [1]. As such, circadian rhythms regulate stem cell fates in many tissues, including the skin, the intestine and the brain [5,6,7].

Neural stem cells (NSCs) are the progenitors of the adult brain and have the ability to generate neurons, astrocytes and oligodendrocytes or to self-renew [8,9]. NSCs reside in mainly two locations in the adult brain: the subventricular zone of the lateral ventricles and the subgranular zone of the dentate gyrus [10]. A majority of NSCs are dormant, or quiescent, existing in a state outside of the cell cycle, which minimizes accumulation of DNA damage during life and allows for long-lived populations [11,12]. Following activation, NSCs enter the cell cycle and give rise to progenitors, which then leave the cell cycle to differentiate into committed cell types. Although NSCs are able to return to quiescence [13], activation of quiescent NSCs results in the gradual decline of NSC populations during aging [14,15]. A disturbed circadian clock results in increased cell cycle entry of hippocampal NSCs, which ultimately results in exhaustion of the NSC population [7]. For instance, loss of circadian clock components *Per2*, *Bmal1 or Rev-Erbα* affected circadian proliferation of NSCs and resulted in increased cell cycle entry of NSCs [16,17,18]. Similarly, loss of transcription factor FoxO3 disrupts quiescence and results in a decrease in the number of NSCs [19,20]. FoxO3 is a transcription factor belonging to the FoxO family, which also includes mammalian family members FoxO1, FoxO4 and FoxO6, and controls expression of many genes involved in the cell cycle and metabolic regulation [21,22]. Moreover, FoxO3 is regulated by the energy-sensing insulin/IGF-1 signalling pathway [23]. Activation of this pathway results in phosphorylation and subsequent nuclear export of FoxO3, inhibiting FoxO3-dependent transcription [24,25].

Previous studies in our lab demonstrated FoxO3 to regulate hepatic circadian rhythms by direct transcriptional control of *Clock* expression [26]. Here, we show that FoxO3 modulates circadian rhythms of neural stem cells through regulation of Clock. Our findings show that FoxO3 is necessary for correct timing of adult neural stem cell proliferation.

## 2. Results

### 2.1. FoxO3 Regulates Circadian Rhythmicity of Neural Stem Cell Cultures via Clock

In the liver, FoxO3 is an important regulator of circadian rhythmicity [26]. In order to examine whether FoxO transcription factors modulate circadian rhythms in neural stem cells, we used an in vitro neural stem cell model known as multipotent astrocytic stem cells (MASCs), which can be grown in monolayers [27,28]. In these cells, we co-transfected a mBmal1::luciferase reporter construct expressing either FOXO1-GFP, FOXO3-GFP or FOXO6-GFP (Figure 1A) or in combination with either *FoxO1* siRNA, *FoxO3* siRNA or *FoxO6* siRNA (Figure 1B) and measured bioluminescence over multiple days in synchronised cultures. We did not interrogate FoxO4 as its expression in the brain is low [29,30]. Overexpression of all FOXO-GFP fusion constructs reduced the amplitude of circadian oscillations in mBmal1::luciferase, when compared to the empty vector control (Figure 1A,C). Importantly, only knockdown *FoxO3* completely abrogated circadian rhythms in mBmal1::luciferase bioluminescence, while knockdown of either *FoxO1* or *FoxO6* reduced the amplitude of oscillation (Figure 1B,D). Quantification of expression levels is shown in Appendix A.

As this is in line with the effects of a loss of FoxO3 in the liver, where it directly regulates *Clock* expression [26], we set out to determine whether this is true for neural stem cells as well. To this end, we transiently downregulated *FoxO3* using siRNA in MASCs and assessed *Clock* expression 24 and 48 h after synchronization (Figure 2A). This showed a clear and persisting downregulation of *Clock* expression for at least two days. We then assessed expression of *Clock* and *FoxO3* over the course of 1 day, starting 24 h after synchronization (Figure 2B,C). This confirmed downregulation of both *Clock* and *FoxO3* expression. In order to determine whether restoration of *Clock* expression could rescue the *FoxO3* knockdown phenotype, as we showed before in NIH3T3 cells, we co-transfected a mBmal1::luciferase reporter construct with either siRNA control and empty vector (Figure 2D, light blue), siRNA control and pFLAG-CLOCK (Figure 2D, dark blue), *FoxO3* siRNA and empty vector (Figure 2E, red) or *FoxO3* siRNA and pFLAG-CLOCK (Figure 2E, orange) and assessed circadian rhythmicity over multiple days. Whereas Clock overexpression did not affect circadian oscillations (Figure 2D,F), it did partially rescue the loss of circadian rhythmicity following knockdown of FoxO3 (Figure 2E,F).

### 2.2. FoxO3 Is Not Required for Circadian Rhythmicity at the Single Cell Level

Recording circadian bioluminescence requires synchronization of cellular clocks within cell cultures [31]. As such, knockdown of *FoxO3* could disrupt circadian synchronization rather than rhythmicity itself. To investigate this, we analysed circadian rhythmicity in single cells expressing Rev-Erbα-VNP fusion protein as a fluorescent marker for the circadian clock. We used NIH3T3 stably transfected with Rev-Erbα-VNP [32], as we did not have the MACS counterpart available. As knockdown of *FoxO3* expression in NIH3T3 cells and MASCs resulted in a similar effect on circadian oscillation at the population level, we expect the same holds true at the single cell level.

We transfected NIH3T3^3C^ cells with *FoxO3* siRNA or siRNA control and assessed Rev-Erbα-VNP fluorescence over multiple days (Figure 3). Surprisingly, knockdown of *FoxO3* did result in a clear reduction in fluorescence level **(**Figure 3A,B), but it did not disrupt circadian rhythmicity in single cells (Figure 3A,C). This shows that the molecular oscillator is still functional following *FoxO3* knockdown but also indicates that *FoxO3* regulates expression of circadian clock components, as the mesor (magnitude) of the oscillation is lower upon FoxO3 knockdown. Since the circadian clock and cell cycle are closely connected [33,34], and timing of proliferation of hippocampal neural stem cells is dependent on circadian rhythmicity [17], loss of FoxO3 could have consequences for the coupling of the circadian clock with the cell cycle. To explore whether FoxO3 affects cell cycle progression, we used NIH3T3^3C^ cells transfected with *FoxO3* siRNA or RNA control as, next to expressing Rev-Erbα-VNP, these cells express fluorescent markers for the G1 phase (hCdt1-mKOrange fusion protein) and combined S, G2 and M phases (hGeminin-CFP fusion protein) of the cell cycle [32]. We examined cell cycle progression over multiple days and quantified the length of G1 and S/G2/M phase in *siC* and *siO3* (Figure 3D–G). This revealed no significant changes in S/G2/M phase or total cell cycle and only a slight trend towards an increased G1 phase length following *FoxO3* knockdown (Figure 3G). Additionally, *FoxO3* knockdown did not disrupt coupling of the circadian clock to the cell cycle as the distribution of circadian cycle length and cell cycle length was similar between *siC* and *siO3* cells.

### 2.3. FoxO3 Is a Clock-Controlled Gene

Previous studies showed circadian expression of FoxO3 in the liver [26,35], suggesting that FoxO3 is a clock-controlled gene. To determine whether expression of FoxO3 is directly regulated by CLOCK–BMAL1, we first searched for occupancy of circadian clock components within gene regulatory regions of *FoxO1*, *FoxO3*, *FoxO4* and *FoxO6* (Appendix A). Since an extensive analysis of genome-wide circadian clock occupancy in neural stem cells is currently unavailable, we analysed publicly available ChIP-seq data for circadian clock components Bmal1, Clock, Per1, Per2, Npas2, Cry1 and Cry2 in the liver [36]. As both circadian oscillations of *FoxO3* expression and a functional role for FoxO3 in circadian rhythms have been established in the liver [26,35], this dataset is valuable in understanding circadian regulation of *FoxO3* expression. Our analysis revealed occupancy of circadian clock components on *FoxO1* and *FoxO3* (Appendix A) but not on *FoxO4* or *FoxO6* (Appendix A). Importantly, multiple regions within the *FoxO3* locus showed occupancy of circadian clock components at the same genomic positions. This was not the case for FoxO1. Since circadian clock components form regulatory complexes consisting of not only CLOCK:BMAL1 heterodimers but also large quaternary CLOCK:BMAL1/PER:CRY structures [4], co-occupancy by multiple circadian clock components is highly indicative of circadian regulation. This supports the notion that FoxO3 is directly transcriptionally regulated by circadian clock components, more so than other FoxO isoforms.

Next, we examined whether circadian clock components bind to the *FoxO3* gene in a circadian fashion (Figure 4 and Appendix A). The core circadian transcriptional system involves rhythmic binding of CLOCK and BMAL1 to regulatory regions within clock-controlled genes [4]. Similarly, we found circadian rhythmicity in the binding of many circadian clock components to multiple regions within the *FoxO3* gene, often in common gene regulatory regions, such as the promoter and first introns, and close to the transcriptional start site (TSS). We also identified rhythmic binding of co-activator CBP, which coincides with circadian *FoxO3* mRNA expression in the liver [35]. Interestingly, this correlated with dynamic histone modifications H3K9ac, H3K27ac, H3K4me3, which are associated with circadian transcription within the *FoxO3* promoter regions [35,37]. The above suggests a coordinated regulation of *FoxO3* by multiple circadian clock components, transcriptional activators and RNA polymerase II. We then explored the circadian landscape of the *FoxO3* gene in detail (Figure 4) and examined two apparent gene regulatory sites: the second intron (Figure 4B) and the promoter (Figure 4C). Importantly, the circadian phase distributions of circadian clock components and transcriptional regulators occupying the *FoxO3* gene closely followed the genome-wide distribution of these components in a circadian cycle, as reported by [36]. This circadian genomic landscape is characterized by, first, a transcriptionally poised state with CLOCK:BMAL1 and NPAS2:BMAL1 occupying, respectively, the FoxO3 promoter and second intron while still being bound to CRY1. As occupancy of CRY1 declines, CLOCK:BMAL1 and NPAS2:BMAL1 are activated and coactivators are recruited to the FoxO3 promoter to the second intron. This is followed by nascent transcription of FoxO3 mRNA, which peaks around the same time of day as the genome-wide de novo transcription in the liver at CT15 [36]. Finally, a repressed transcriptional state is created when CLOCK:BMAL1 and NPAS2:BMAL1 occupancy decline and the occupancy of repressors PER1, PER2 and CRY2 occupancy increases. Together, these results suggest that *FoxO3* is a clock-controlled gene, directly regulated by CLOCK–BMAL1. This is important because it places FoxO3 in a central position, regulating and simultaneously being regulated by circadian rhythms.

### 2.4. FoxO3 Modulates Circadian Proliferation of Hippocampal Neural Stem Cells

As was shown before by Bouchard-Cannon and coworkers, NSC proliferation in the adult hippocampus is subject to diurnal rhythmicity [7,17]. In order to study if the FoxO3 driven modulation of circadian rhythms in neural stem cells in culture is possibly translated into a role in adult hippocampal neural stem cells, we set out to measure diurnal NSC proliferation in vivo WT and FoxO3^−/−^ mice. To this end, we isolated brains of 6-month-old FoxO3^−/−^ and FoxO3^+/+^ mice [38] every four hours during a circadian cycle. We then assessed diurnal NSC proliferation by determining the number of Ki67-positive cells in the subgranular zone of the dentate gyrus in the hippocampi of FoxO3^−/−^ and FoxO3^+/+^ mice at each time point (Figure 5A,B). This revealed a clear 24 h rhythm in proliferation of SGZ NSCs in FoxO3^+/+^ mice peaking at the end of the light phase (~ZT11, ANOVA: *F =* 17.45, *p* < 0.001; CIRCWAVE: r^2^ = 0.85, *p* < 0.0001, Figure 5C). This is in line with a previously reported diurnal proliferation of SGZ NSCs in wild-type mice, which peaked at ZT12-15 [17]. The small difference in peak time could be due to the use of BrdU by Bouchard-Cannon and colleagues as a proliferation marker rather than Ki67. As BrdU needs time to be incorporated into the DNA of dividing cells before it can be detected as a proliferation marker [39], a delay is expected.

In FoxO3^−/−^ mice, a 24 h oscillation in NSC proliferation was also present but peaking at the early dark phase (~ZT14, ANOVA: *F* = 4.58, *p* < 0.05; CIRCWAVE, r^2^ = 0.72, *p* < 0.01; Figure 5C). This phase delay is significantly different, as assessed using CircaCompare (Figure 5D) [40]. As such, the diurnal oscillation in proliferation of FoxO3-deficient NSCs was phase-delayed by almost three hours compared to the diurnal rhythm in proliferation exhibited by wild-type NSCs. In addition, the total proliferation over 24 h was clearly reduced in FoxO3^−/−^ mice when compared to FoxO3^+/+^ controls (Figure 5D). This was expected as FoxO3^−/−^ mice show a reduced number of SGZ NSCs at 6-months old [19]. These results suggest that FoxO3 is a modulator of circadian dynamics in NSC fates and necessary for the correct timing of hippocampal proliferation.

## 3. Discussion

Neural stem cell maintenance is required for lifelong neurogenesis and governed by circadian rhythms [7,41]. FoxO3 is an important regulator of neural stem cell fates [19,20], as well as circadian rhythms in the liver [26]. Here, we show that loss of *FoxO3* disrupts circadian rhythms in neural stem cell cultures. Moreover, FoxO3 knockdown led to a clear downregulation of *Clock* expression, regardless of circadian time. *Clock* expression, in turn, could partially rescue this phenotype. These findings are similar to the effects observed after downregulation of *FoxO3* in mouse liver cells or embryonic fibroblasts [26], suggesting a universal function of FoxO3 in circadian regulation across different cell types. To determine whether FoxO3 is required for circadian rhythmicity or for circadian synchronization within cultured cells—loss of either could explain the disruption of circadian rhythms in bulk recordings—we examined circadian rhythmicity at the single cell level in NIH3T3^3C^ cells containing a Rev-Erbα-VNP fluorescence marker [32]. Surprisingly, we still observed circadian rhythms in cells with *FoxO3* siRNA, with no significant changes in amplitude. This demonstrates that FoxO3 it not required for circadian rhythmicity but is likely necessary for synchronization between different cells. As a result, downregulation of *FoxO3* effectively eliminates circadian oscillations in a bulk recording.

Importantly, dexamethasone, used here to synchronize circadian rhythms in NSCs, is a glucocorticoid that regulates *FoxO3* expression via glucocorticoid response elements within the FoxO3 promoter [42]. Whereas ablation of FoxO3 disrupts circadian rhythmicity of NIH3T3 cells synchronized with forskolin as well [26], forskolin-dependent synchronization relies on increased cAMP levels [43]. Increased cAMP levels, in turn, activate AMPK-FoxO3 signalling, thereby promoting FoxO3 gene expression and transcriptional activity [42,44]. Since both glucocorticoid and cAMP signalling are important for synchronization of circadian oscillators [43,45], FoxO3 could be a key downstream effector of these pathways. Interestingly, circadian glucocorticoid oscillations help to preserve quiescent NSCs in the adult hippocampus [46]. As such, FoxO3-dependent quiescence of NSCs could be modulated by glucocorticoid signalling.

Interestingly, loss of FoxO3 clearly reduced Rev-Erbα-VNP fluorescence levels in single NIH3T3^3C^ cells. This is in line with downregulation of *Rev-Erbα (Nrdr1)* expression following FoxO3 knockdown in NIH3T3 cells [26]. This could be the result of reduced *Clock* expression, as CLOCK–BMAL1 regulates *Rev-Erbα* expression [3,4]. However, this could also be a direct transcriptional regulation by FoxO3 as we identified FoxO3 binding sites in the promoter region of *Rev-Erbα* in NSCs [47]. Together, these findings suggest that FoxO3, although not required for circadian rhythmicity, does affect the molecular oscillator by transcriptionally regulating the expression of various components, including *Clock*, *Nrdr1* and *Bmal1*.

This is interesting as we found various circadian clock components binding to the sites within the *FoxO3* promoter and its second intron in publicly available ChIP-seq data in the liver [36]. Not only did circadian clock components bind to *FoxO3*, their binding sites were multiple and overlapping, suggesting the presence of CLOCK–BMAL1 and CLOCK:BMAL1/PER:CRY regulatory complexes [3,4]. Moreover, their proximity to binding sites of transcriptional regulators CBP and p300, as well as to binding of RNA polymerase II, suggests a direct transcriptional regulation. Importantly, the occupancy of these circadian clock components was dynamic and followed a circadian rhythm with phase-distributions of activators and repressors closely resembling their genome-wide circadian landscape [36]. Whereas CLOCK and BMAL1 bound to the *FoxO3* promoter region and peaked in their circadian occupancy at the same time, this was not the case for the second intron where BMAL1 and CLOCK occupancy peaked at different times of the day. In the second intron, however, BMAL1 and NPAS2 shared circadian occupancy at the same time of day. Since NPAS2 is the preferred heterodimeric partner of BMAL1 in the forebrain [48], it would be interesting to examine circadian regulation of *FoxO3* expression in different cell types.

As FoxO3 regulates NSC proliferation [19], which is controlled by the circadian clock [7], we investigated the consequences of a loss of FoxO3 on diurnal rhythmicity in NSC proliferation. Although FoxO3-deficient NSC proliferation adhered to a 24 h rhythm, this oscillation was, surprisingly, phase-delayed compared to wild-type controls. This indicates that FoxO3 is not required for diurnal rhythmicity of NSC proliferation in contrast to mutant mice lacking Bmal1, Per2 and Rev-Erbα [16,17,18]. Instead, FoxO3 appears to coordinate timing of proliferation, likely by controlling cell cycle entry and exit. In line with the above, it would be interesting to determine the cellular consequences of the phase-shift in diurnal hippocampal proliferation. Circadian regulation of cell division is thought to segregate DNA replication away from periods of high metabolic activity as these processes produce mutation-causing free radicals [49,50]. Circadian disturbance, characteristic of shift work and sleep disruption, is associated with metabolic pathologies and cancer [51,52]. Since we found the peak in FoxO3-deficient NSC proliferation to be considerably delayed, it is conceivable that FoxO3-deficient NSCs are exposed to increased levels of oxidative stress, possibly resulting in increased cell death, senescence, stem cell dysfunction or an accumulation of mutations in their progeny. In the case of SVZ NSCs, such an increase in mutations could lead to a higher frequency of brain tumours [53].

FoxO3 is a key regulator of aging [54] and its targets change with age [55]. Similarly, the circadian transcriptome has been found to change during aging, in the liver and in epidermal and muscle stem cells [56]. One possibility is that FoxO3 could be necessary to modulate circadian function throughout life. Interestingly, aging results in desynchrony of SCN neurons [57] and reduces the robustness of synchronizing cues to peripheral tissues [58]. The loss of circadian rhythmicity in NSC cultures but not in single cells following knockdown of *FoxO3* indicates impaired synchronization. Therefore, FoxO3 could be an important regulator of circadian synchronization during aging to promote cellular homeostasis in response to circadian and metabolic challenges. This is interesting as there is evidence of circadian heterogeneity in populations of hair-follicle stem cells and epidermal stem cells [6,59]. This affects stem cell function as it creates a heterogeneous stem cell population with individual stem cells differentially responding to cues promoting activation or dormancy. FoxO3 could have a role in balancing such a circadian stem cell heterogeneity with timely responses to circadian cues of nutrients and cellular stress, helping to preserve NSC populations throughout life.

Importantly, we found extensive occupancy of circadian clock components on FoxO3 but not of FoxO1, FoxO4 or FoxO6 whereas only knockdown of *FoxO3* resulted in complete abrogation of circadian rhythmicity. This indicates an exclusive role for FoxO3 in circadian regulation. Although all FoxOs recognize the same DNA motifs or DBEs and are partially redundant, FoxOs do show functional diversification [38]. As the effects of FoxO3 appear to be similar across cell types (in fibroblasts, hepatocytes [26] and neural stem cells (this study)), it would be interesting to find FoxO3-specific targets involved in circadian rhythms.

To conclude, our results demonstrate that FoxO3 is a regulator of circadian rhythms in NSCs and is itself also regulated by the circadian clock. We suggest that FoxO3 is necessary for the correct timing of hippocampal NSC proliferation to ensure its alignment with cellular metabolism. This could offer valuable insights in understanding the relationship between circadian rhythms, aging, cancer and other pathologies.

## 4. Materials and Methods

### 4.1. FoxO3 Mutant Mice

All experiments involving animals and their care were performed in accordance with the guidelines of the University of Amsterdam, national guidelines and laws. This study was approved by the Dutch Animal Ethics Committee. Mice were housed in standard cages under a 12 h light/dark cycle, with ad libitum access to food and water. Wild type mice from strain C57BL6 were used. FoxO3 mutant mice [38] were used in heterozygous breeding generating wild-type (^+/+^) and mutant progeny (^−/−^). All experimental and control mice were littermates. Both male and female mice were used for all in vivo genetic studies.

### 4.2. Genotyping

For genotyping FoxO3 null and wild-type sibling mice, three primers were used in the same PCR reaction: Forward primer 1: 5′-ATT CCT TTG GAA ATC AAC AAA ACT-3′; Reverse primer 1: 5′-TGC TTT GAT ACT ATT CCA CAA ACC C-3′; Reverse 2: 5′-AGA TTT ATG TTC CCA CTT GCT TCC T-3′. FoxO3 wild type allele produced a band of 100 bp and the FoxO3 null allele produced a band of 186 bp in a Taq DNA polymerase PCR reaction.

### 4.3. Tissue Preparation

Mice were euthanized every 4 h starting at CT2 (10:00 AM) and brains were fixed in 4% PFA for 8 h at 4 °C and then in 30% sucrose in PBS overnight at 4 °C for immunohistochemistry. Coronal and sagittal sections (16 μm) were cut using a cryostat (Leica, Wetzlar, Germany) and mounted on slides.

### 4.4. Immunohistochemistry

Immunohistochemistry was performed using primary antibodies rabbit anti-Ki67 (1:500, Abcam, Cambridge, UK, ab15580) using heat-induced antigen retrieval. Briefly, sections were incubated in 10 mM sodium citrate (pH 6.0) for 15 min at 90 °C. Sections were then blocked with 4% hiFBS (Gibco, Waltham, MA, USA) in a Tris-buffered saline solution pH 7.6 containing Triton X-100 and incubated with primary antibody overnight at 4 °C. After three washes with a Tris-buffered saline solution, sections were incubated with secondary antibodies donkey anti-rabbit IgG Alexa Fluor 488 or 594 or goat anti-mouse IgG Alexa Fluor 555 (Molecular Probes, Eugene, OR, USA) at 1:1000 for 2 h at room temperature. Sections were counterstained with DAPI and embedded in FluorSave (Sigma, St. Louis, MO, USA).

### 4.5. Image Acquisition and Quantification

All fluorescent images were taken with a Leica microscope (DFC310FX). All quantifications were performed by an observer blinded to the experimental condition. Images of multiple sections were quantified and averaged per animal. The sections spanned the rostrocaudal length of the hippocampus. Only DAPI+ cells were included. Quantifications were performed in the subgranular zone of the dentate gyrus only.

### 4.6. Cell Culture

Multipotent astrocytic stem cells (MASCs) were cultured in DMEM F-12 (Invitrogen, Waltham, MA, USA) + 10% hiFBS (Gibco, Mississauga, ON, Canada), 200 nM L-Glutamine (Gibco), 1× N2 (Invitrogen, Waltham, MA, USA), 1× Penicillin-Streptomycin (Gibco, Waltham, MA, USA) and supplemented with bFGF (10 ng/mL, Sigma, St. Louis, MO, USA), EGF (10 ng/mL, Sigma, St. Louis, MO, USA) and BPE (12.5 mg/mL, Invitrogen, Waltham, MA, USA). Cells were incubated at 37 °C, 5% CO_2_. MASCs were a gift of C. Fitzsimons. For luciferase assays, Neuro2a cells were cultured in DMEM (Invitrogen, Waltham, MA, USA) supplemented with L-glutamine, 10% FBS (Gibco) and 1% penicillin/streptomycin (Gibco) at 37 °C in 5% CO_2_. For single cell fluorescence microscopy, NIH3T3^3c^ cells containing *Rev-Erbα-VNP* clock reporter and FUCCI *hCdt1-mKOrange* and *hGeminin-CFP* FUCCI cell cycle reporter genes [32] were cultured in DMEM/F10 containing 10% FBS (Gibco), 1× Penicillin–Streptomycin at 37 °C and 5% CO_2_ (pH7.7).

### 4.7. Plasmids, siRNA Oligos and Transfection

For overexpression studies, the following plasmids (600 ng per transfection) were used: pFOXO1-EGFP-N1, pFOXO3-EGFP-N1, pFOXO6-EGFP-N1 and p-EGFP-N1 as empty vector control. pGL4.11-Bmal1::luciferase (200 ng) was used as circadian reporter, pFlag-CLOCK (600 ng) was used to rescue FoxO3 siRNA with pcDNA-Neo as empty vector control. For FoxO3 promoter studies, two plasmids with 1.4 kb and 2.1 kb of the *FoxO3* promoter cloned into a pGL3::luciferase vector (see below) were used with pGL3::luciferase as empty vector control. For knockdown studies, siRNA oligos (50 pmol) against FoxO1 (siO1), FoxO3 (siO3), FoxO6 (siO6) or a scrambled siRNA (siC) were used. Transient transfections were performed using Lipofectamine2000 (Invitrogen, Waltham, MA, USA) according to the manufacturer’s instructions unless specified otherwise.

### 4.8. Real-Time Bioluminescence Recordings

To monitor circadian oscillations in MASCs, cells were plated in 35 mm dishes and transfected the following day. Forty-eight hours later, the medium was replaced with medium containing 25 mM HEPES and 0.1 mM luciferin (Sigma, St. Louis, MO, USA) and 100 nM dexamethasone for circadian synchronization. Bioluminescence was then recorded for multiple days (60 sec measurements at 10 min intervals) with a LumiCycle 32-channel automated luminometer (Actimetrics, Wilmette, IL, USA) placed in a dry, temperature-controlled incubator at 37 °C. Data were analysed with the Actimetrics software to calculate circadian amplitude.

### 4.9. RNA Isolation and Quantitative Real-Time PCR

RNA was harvested 24 and 48 h after transfection using TriZOL (ThermoFisher, Waltham, MA, USA) according to the manufacturer’s instructions. Relative expression levels were determined by quantitative real-time PCR (Lightcycler 480) using the QuantiTect SYBR Green RT PCR Kit (QIAGEN, Hilden, Germany) according to the manufacturer’s instructions. Gene expression was calculated relative to endogenous controls B2M, and normalised to the expression of control samples in each group, to give a ΔΔCt value. Primers used were: *B2m* forward: 5′-TTC TGG TGC TTG TCT CAC TG-5′, *B2m* reverse: 5′-CAG TAT GTT CGG CTT CCC ATT C-3′, *FoxO3* forward: 5′-CCT ATG CCG ACC TGA TCA CC-3′, *FoxO3* reverse: 5′-ATT CTG AAC GCG CAT GAA GC’-3′, *Clock* forward: 5′-AGG CTA TTT GCC ATT TGA AGT CT-3′ and *Clock* reverse: 5′-GCT CGT GAC ATT TTG CCA GAT TT-3′.

### 4.10. Time-Lapse Fluorescence Microscopy

NIH3T3^3c^ cells were plated in 4 well poly-L-lysine coated glass bottom dishes (D141410, Matsunami Glass Ind.) and reverse transfection was performed using Lipofectamine^®^ RNAiMAX (Invitrogen) with *FoxO3* siRNA, *FoxO6* siRNA or siRNA control. After 24 h, transfection medium was replaced with regular culture medium and placed in a temperature and humidity-controlled chamber (37 °C, 5% CO_2_) of a live cell imaging Zeiss LSM510/Axiovert 200M confocal microscope. Images were then recorded every 30 min for 72 h or more using a Coolsnap HQ/Andor Neo sCMOS camera. Live cell imaging was conducted and analysed as described previously [32,60]. Briefly, time series for each of the fluorescent markers (Rev-ERbα-VNP, hCDT1-mKOrange, hGeminin-CFP) were generated and used to assess lengths of the circadian cycle, total cell cycle, G1 phase and combined S, G2 and M phases (S/G2/M). The G1 phase was defined as the interval between the peaks of hGeminin-CFP and hCDT1-mKOrange expression whereas the S/G2/M phase was defined as the interval between the peaks of hCDT1-mKOrange and hGeminin-CFP expression.

### 4.11. Statistical Analysis

All error bars represent the mean ± SEM. Significance is stated as follows: *p* > 0.05 (ns), *p* < 0.05 (*), *p* < 0.01 (**), *p* < 0.001 (***), *p* < 0.0001 (****). Statistical details of each experiment can be found in the figure legend. n represents number of animals in vivo or independent biological repeats in vitro. Statistical analysis of circadian oscillation was performed with Circwave (courtesy of Prof. Roelof Hut, http://www.euclock.org/, version 1.4) and CircaCompare [40].

## Figures and Tables

**Figure 1 ijms-24-13662-f001:**
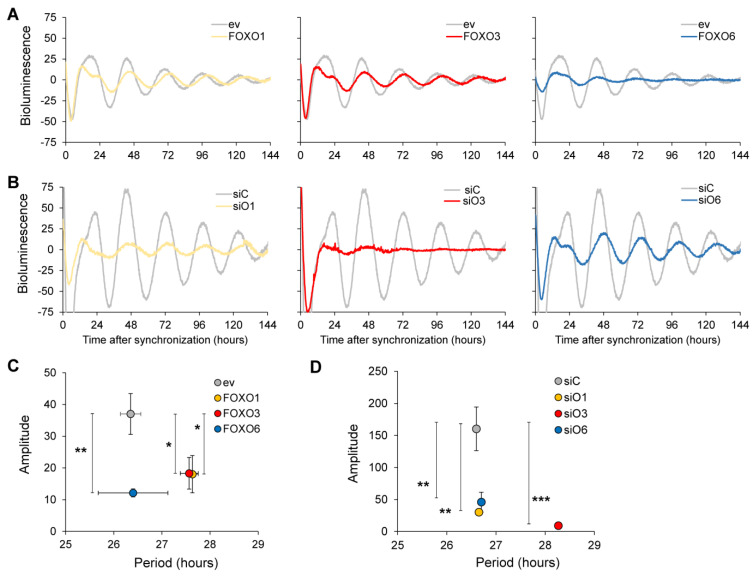
Differential effect of FoxO transcription factors on circadian oscillations in neural stem cells in vitro. (**A**) Representative examples of bioluminescence rhythms in cells co-transfected with a mBmal1::luciferase reporter construct and either empty vector (grey line), FOXO1-EGFP (left, yellow line), FOXO3-EGFP (middle, red line), or FOXO6-EGFP (right, blue line). (**B**) Representative examples of bioluminescence rhythms in cells co-transfected with a mBmal1::luciferase reporter construct and control siRNA (grey line), *FoxO1* siRNA (left, yellow line), *FoxO3* siRNA (middle, red line), or *FoxO6* siRNA (right, blue line). (**C**,**D**) Graphical representation of the amplitude and period length under FOXO overexpression (**C**) or *FoxO* knockdown (**D**). Values are averages of independent experiments (n = 3) performed in triplicate. Error bars represent SEM. One-way ANOVA with Dunnett’s multiple comparison test. * *p* < 0.05; ** *p* < 0.01; *** *p* < 0.001.

**Figure 2 ijms-24-13662-f002:**
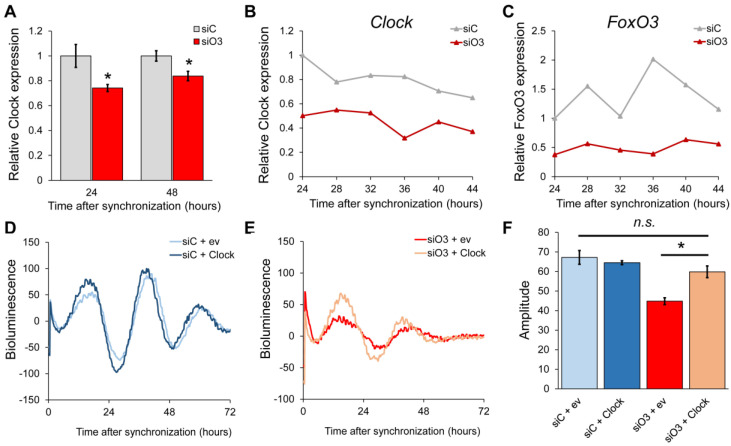
*Clock* rescues the FoxO3 siRNA phenotype in neural stem cells. (**A**) *Clock* mRNA expression in adult neural stem cells following 24 and 48 h after transfection of siRNA control or *FoxO3* siRNA. (**B**) *Clock* expression in adult neural stem cells following siRNA control or *FoxO3* siRNA during 24 h following circadian synchronization with dexamethasone. (**C**) *FoxO3* expression in adult neural stem cells following siRNA control or *FoxO3* siRNA during 24 h following circadian synchronization with dexamethasone. (**D**,**E**) Representative examples of bioluminescence rhythms in cells co-transfected with a mBmal1::luciferase reporter construct and either siRNA control plus empty vector (light blue), siRNA control plus pFlag-CLOCK (dark blue), *FoxO3* siRNA plus empty vector (red) or *FoxO3* siRNA plus pFlag-CLOCK (orange). (**F**) Quantification of amplitudes of bioluminescence rhythms shown in D and E. Values are averages of biological replicates (n = 3) performed in triplicate. One-way ANOVA with Tukey’s multiple comparison test. Error bars represent SD. * *p* < 0.05, n.s. is “not significant”.

**Figure 3 ijms-24-13662-f003:**
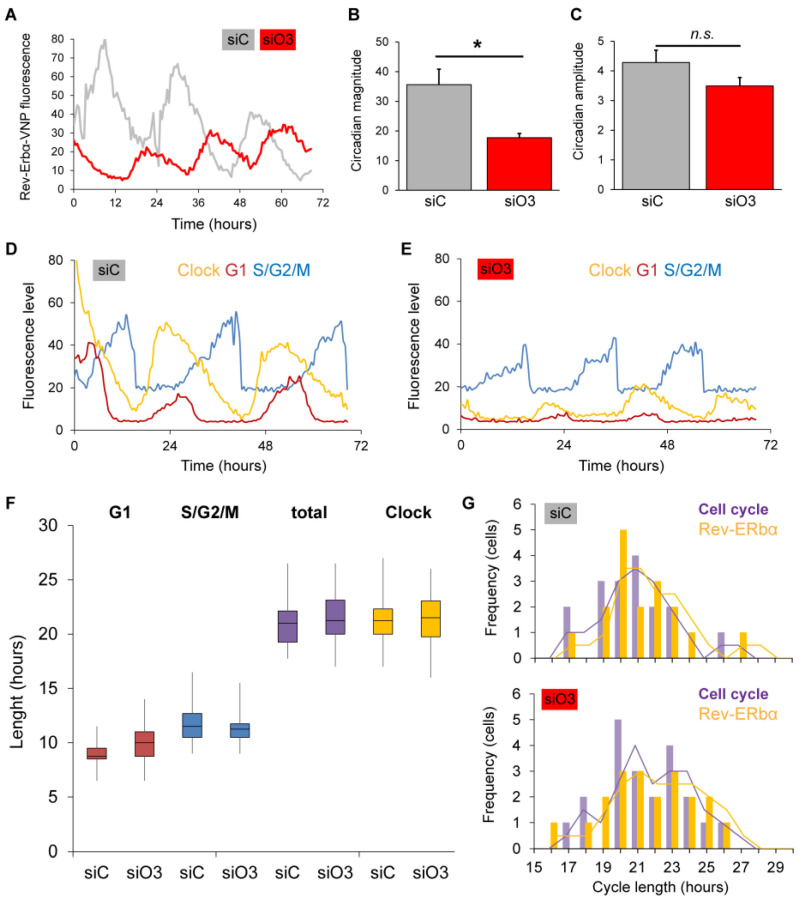
Loss of *FoxO3* does not disrupt circadian rhythmicity in single cells. (**A**) Single cell recordings over multiple days of Rev-ERbα-VNP fluorescence levels in NIH3T3^3C^ transfected with *FoxO3* siRNA (siO3, red) or control siRNA (siC, grey). Representative examples. (**B**) Magnitude of circadian oscillations (mesor) in Rev-ERbα-VNP fluorescence levels shown in (**A**). siC, n = 13; siO3, n = 20. * *p* < 0.025. (**C**) Amplitude of circadian oscillation in Rev-ERbα-VNP fluorescence levels shown in (**A**). siC, n = 13; siO3, n = 20. n.s., not significant. (**D**,**E**) Representative examples of single cell recordings of fluorescence marker expression for circadian clock oscillations (Rev-ERbα-VNP, yellow), G1 phase of the cell cycle (hCdt1-mKOrange fusion protein, red) and combined S/G2/M phases (hGeminin-CFP fusion protein, blue) in NIH3T3^3C^ cells transfected with *FoxO3* siRNA (**D**) or siRNA control (**E**). (**F**) Comparison of the average length of cell cycle phase S/G2/M (blue), G1 (red) and circadian cycle (yellow) of data shown in (**D**–**G**). siC, n = 13; siO3, n = 20. n.s., not significant. (**G**) Frequency histograms of circadian cycle length (Clock, yellow) and total cell cycle length (purple) in single NIH3T3^3C^ cells transfected with *FoxO3* siRNA (bottom) or siRNA control (top). siC, n = 13; siO3, n = 20.

**Figure 4 ijms-24-13662-f004:**
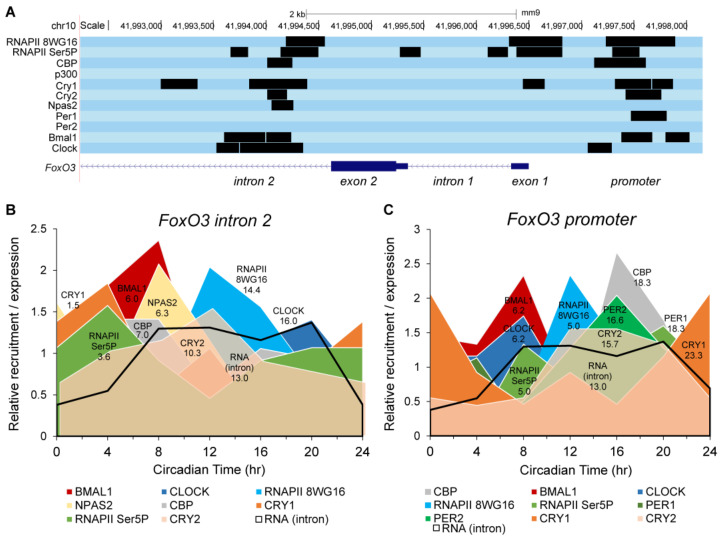
Circadian landscape of the *FoxO3* gene. (**A**) University of California Santa Cruz (UCSC) genome browser track view of the *FoxO3* gene showing ChIP-seq binding peaks for circadian clock components Clock, Bmal1, Per1, Per2, Npas2, Cry1 and Cry2, as well as transcriptional regulators CBP, p300 and recruitment and initiation of RNA polymerase II (8WG16 and Ser5P, respectively). Data from [36]. (**B**,**C**) Circadian landscape of the second intron (**B**) and promoter (**C**) within the *FoxO3* gene showing phase distributions of Bmal1, Clock, Per1, Per2, Npas2, Cry1 and Cry2, CBP and RNAPII recruitment and initiation. The mean circadian phase of peak binding is indicated under the name. The black line denotes circadian expression of *FoxO3* mRNA.

**Figure 5 ijms-24-13662-f005:**
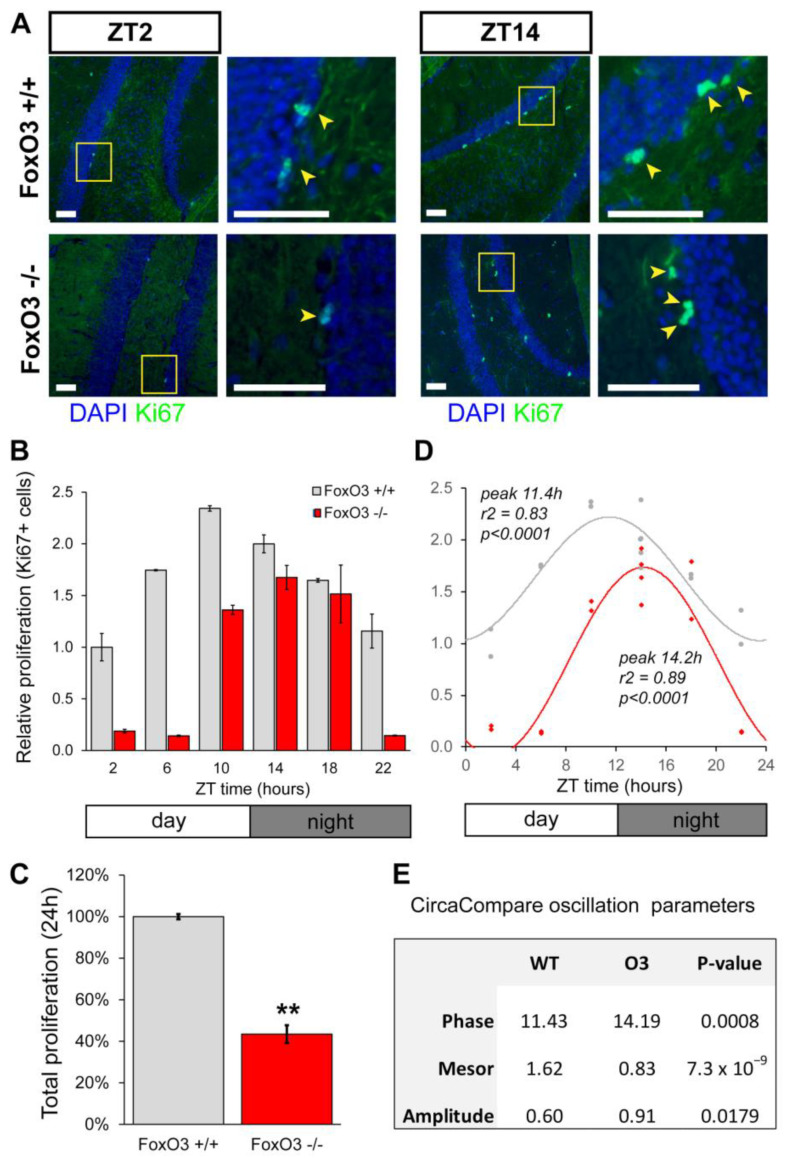
Diurnal proliferation of neural stem cells in the dentate gyrus of adult FOXO3-deficient mice. (**A**) Representative examples of Ki67-positive cells (yellow arrow heads) in the subgranular zone of the dentate gyrus of FoxO3^−/−^ mice and littermate controls at CT2 and CT14. (**B**) Diurnal proliferation of neural stem cells in the dentate gyrus of 6-month-old FoxO3^+/+^ (grey) and FoxO3^−/−^ mice (red). Values are averages per genotype and normalized to CT2 of controls. Error bars represent SEM. (**C**) Data in (**B**) shown as total proliferation over 24 h in 6-month-old FoxO3^+/+^ and FoxO3^−/−^ mice (n ≥ 14). Values are average sum of all time points per genotype. Error bars represent SEM. (** *p* < 0.01). (**D**) Data in (**B**) fitted to a linear harmonic regression model to determine circadian rhythmicity (CircWave). (**E**) Parameter comparison of circadian oscillation in Ki67-positive cells in the subgranular zone of the dentate gyrus of FoxO3^+/+^ and FoxO3^−/−^ mice, calculated with CircaCompare [40].

## Data Availability

Data are available upon request.

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
