# Peer review of "FoxO3 Modulates Circadian Rhythms in Neural Stem Cells"

_ijms, 2023, doi:10.3390/ijms241713662_

Round 1
Reviewer 1 Report
1. Overexpression of FOXO6 also significantly reduced the amplitude of circadian oscillations, any hypothesis on this?
2. Any experiment can confirm the overexpression of FOXO-GFP constructs, like westernblot?
3. Please describe the ChIP Seq analysis in details in the method section.
Author Response
We thank the reviewer for the positive comments and suggestions. The response to the comments is shown in blue.
Answer to the Reviewer’s Comments:
- Overexpression of FOXO6 also significantly reduced the amplitude of circadian oscillations, any hypothesis on this?
This is correct, overexpression of FOXO6 also impacts the amplitude of oscillation, as occurs in NIH3T3 cells (as we have shown previously, ref 26). However, it does not affect clock components directly. We are currently further investigating the role of FOXO6 in circadian rhythms and other processes in neural stem cells.
- Any experiment can confirm the overexpression of FOXO-GFP constructs, like westernblot?
We have performed experiments to confirm overexpression and knockdown of the different FoxOs. but had not included them. This has been corrected and we have now included qPCR data on the overexpression and knockdown experiments (Figure S1 of the revised manuscript).
- Please describe the ChIP Seq analysis in details in the method section.
We did not include the ChIP seq analysis in the methods because we preformed a meta-analysis using previously published data (not performed for this study). The ChIP seq data and protocols were published in ref 36.
Reviewer 2 Report
The aim of this paper is to define the role of the FoxO3 transcription factor in circadian rhythmicity of neural stem cells. This brings together two strands from previous work: FoxO3 plays a role in circadian regulation in the liver, and it is important in neural stem cell maintenance. In this paper the authors first show that knocking down FoxO3 in neural stem cells in vitro dampens the oscillation of a circadian reporter gene. Knockdown of FoxO3 in neural stem cells is then shown to correlate with a decrease in Clock gene expression, and co-expressing Clock can partially rescue the damping of the rhythm. The authors then switch to NIH3T3 cells in culture to look at single cell rhythms and cell cycle markers to determine whether FoxO3 knockdown affects individual rhythms or synchronization in a population, and conclude that rhythms and cell cycle progression in single cells are not disrupted. Next the authors analyze published datasets of ChIP-seq results (from liver cells) to look at occupancy of circadian clock proteins on the FoxO3 gene, and map the timeline of circadian transcription factors coming on and off of the locus. Finally, the authors use mice with a FoxO3 knockout to look at proliferation of neural stem cells in vivo and find a reduction and a phase delay in the rhythm of cell division. The authors conclude that FoxO3 is important in regulating circadian rhythms in neural stem cells, and important in maintaining cell populations through the circadian regulation of the cell cycle.
The paper is well-written and understandable. The figures are clearly presented. On the whole the conclusions are supported by the evidence, with the exceptions described below.
The following problems and questions need to be addressed by the authors:
1) Figure 1: It’s hard to say whether FOXO3 is more effective than FOXO1 or 6 at damping the rhythm when there is no assay of the effectiveness of the knockdowns. Are all the siRNAs equally effective at reducing gene expression of their targets?
2) Figure 2: The N is stated as 2. The statistical tests reported in the figure legend are not valid with N less than 3. The authors can report a mean and a range for the two values but should delete the stats and the conclusions based on those stats.
3) Figure 3: I don’t understand how amplitude is measured. It looks to me that amplitude is clearly reduced in the FoxO3 siRNA knockdown cells (Fig 1A). Amplitude is defined by Parsons et al for the CircaCompare software (used by the authors) in the standard way, which is “half of the difference between the peak and trough”.There is a very large difference in amplitude between the two curves shown in Figure 3A.
4) Figure 3: Why did the authors switch to a different cell type (NIH3T3)? This needs to be justified. If the authors want to make conclusions about neural stem cells, can they justify applying findings from NIH3T3 cells to their neural stem cells?
5) Figure 4: The question above regarding NIH3T3 cells also applies to the ChIP-seq data analyzed for Figure 4. Can the authors justify applying data collected in liver cells to their neural stem cells? These two cell types should have different sets of transcription factors and the liver results might not apply to another cell type.
6) Page 6 lines 195-206: This paragraph is out of place. It should be deleted and combined with the following paragraph.
7) Figure 5: These results were collected from animals kept in a light/dark cycle. The graphs are correctly labelled as ZT (zeitgeber time) to indicate this, but the results are described in the text as a “circadian rhythm”. This is strictly not accurate since the rhythm was not measured under constant conditions; this is a diurnal rhythm under a light/dark cycle.
8) Figure 5: Panel letters C and D are swapped to the wrong panels.
What is N for the statistics?
9) Figure S2: Part A is a duplication of Figure S1B and should be deleted.
10) The conclusion that FoxO3 maintains neural stem cell populations through circadian regulation is a bit over-stated. There is no direct evidence that the decrease in NSC numbers in FoxO3 knockout mice is a result of disruption of circadian regulation. There is a correlation, but no direct causal link between circadian disruption and NSC decrease.
Author Response
We thank the reviewer for the positive comments and suggestions. The response to the comments is shown in blue.
Answer to the Reviewer’s Comments:
1) Figure 1: It’s hard to say whether FOXO3 is more effective than FOXO1 or 6 at damping the rhythm when there is no assay of the effectiveness of the knockdowns. Are all the siRNAs equally effective at reducing gene expression of their targets?
Thank you for noticing the missing controls. We have performed these experiments, but did not include them in the manuscript. We have now included them in the revision (figure S1 of the revised manuscript). Knockdown efficiency is similar in all cases.
2) Figure 2: The N is stated as 2. The statistical tests reported in the figure legend are not valid with N less than 3. The authors can report a mean and a range for the two values but should delete the stats and the conclusions based on those stats.
Where it reads N=2 it should be N=3, this was a typing mistake and has been corrected.
3) Figure 3: I don’t understand how amplitude is measured. It looks to me that amplitude is clearly reduced in the FoxO3 siRNA knockdown cells (Fig 1A). Amplitude is defined by Parsons et al for the CircaCompare software (used by the authors) in the standard way, which is “half of the difference between the peak and trough”.There is a very large difference in amplitude between the two curves shown in Figure 3A.
Because there is also a large difference in mesor (magnitude) between siC and siO3 (Figure 3A), we opted to calculate amplitude as peak/trough. This eliminates effects resulting from the difference in mesor. If required we could represent amplitude as “half of the difference between the peak and trough”, and normalize for mesor.
4) Figure 3: Why did the authors switch to a different cell type (NIH3T3)? This needs to be justified. If the authors want to make conclusions about neural stem cells, can they justify applying findings from NIH3T3 cells to their neural stem cells?
We observed that the effects of FoxO3 knockdown and overexpression are comparable between NIH3T3 cells and MASCs. The single cell analysis was not possible in MASCs, and we opted for this analysis in the NIH3T3-3C line, which carries reporter genes for the circadian clock and cell cycle. We have now included this in the revised manuscript.
5) Figure 4: The question above regarding NIH3T3 cells also applies to the ChIP-seq data analyzed for Figure 4. Can the authors justify applying data collected in liver cells to their neural stem cells? These two cell types should have different sets of transcription factors and the liver results might not apply to another cell type.
So far, all effects of FoxO3 knockdown or knockout are comparable between NIH3T3, MASCs and liver. Although these express different sets of transcription factors, circadian oscillator components are expressed in all. We feel confident that this comparison can be made.
6) Page 6 lines 195-206: This paragraph is out of place. It should be deleted and combined with the following paragraph.
This has been corrected.
7) Figure 5: These results were collected from animals kept in a light/dark cycle. The graphs are correctly labelled as ZT (zeitgeber time) to indicate this, but the results are described in the text as a “circadian rhythm”. This is strictly not accurate since the rhythm was not measured under constant conditions; this is a diurnal rhythm under a light/dark cycle.
We thank the reviewer for noticing this. It is true that in entrained conditions one cannot talk about circadian rhythms. This has been adjusted as suggested.
8) Figure 5: Panel letters C and D are swapped to the wrong panels.
Thank you for noticing, this has been corrected.
What is N for the statistics?
Regarding the Circwave and Circacompare (panels D ann E), the calculations take all points into account (at least 2 animals per time point). Regarding panel C, N>14.
9) Figure S2: Part A is a duplication of Figure S1B and should be deleted.
We have deleted panel S2A.
10) The conclusion that FoxO3 maintains neural stem cell populations through circadian regulation is a bit over-stated. There is no direct evidence that the decrease in NSC numbers in FoxO3 knockout mice is a result of disruption of circadian regulation. There is a correlation, but no direct causal link between circadian disruption and NSC decrease.
We agree that there is no direct evidence. We therefore removed this conclusion from the abstract, and added a tuned-down version to the discussion.